# Performance of 16S rRNA Gene Next-Generation Sequencing and the Culture Method in the Detection of Bacteria in Clinical Specimens

**DOI:** 10.3390/diagnostics14131318

**Published:** 2024-06-21

**Authors:** Alexandru Botan, Giuseppina Campisciano, Verena Zerbato, Stefano Di Bella, Omar Simonetti, Marina Busetti, Dan Alexandru Toc, Roberto Luzzati, Manola Comar

**Affiliations:** 1Faculty of Medicine, “Iuliu Hațieganu” University of Medicine and Pharmacy, 400012 Cluj-Napoca, Romania; botan.alexandru@elearn.umfcluj.ro; 2Laboratory of Advanced Microbiology Diagnosis and Translational Research, Institute for Maternal and Child Health IRCCS Burlo Garofolo, 34137 Trieste, Italy; giuseppina.campisciano@burlo.trieste.it (G.C.);; 3Clinical Department of Medical, Surgical and Health Sciences, Trieste University, 34129 Trieste, Italy; 4Infectious Diseases Unit, Trieste University Hospital, 34125 Trieste, Italy; 5Microbiology Unit, Trieste University Hospital (ASUGI), 34125 Trieste, Italy; 6Department of Microbiology, “Iuliu Hațieganu” University of Medicine and Pharmacy, 400012 Cluj-Napoca, Romania

**Keywords:** 16S NGS, diagnosis, culture-based diagnosis, diagnostic microbiology

## Abstract

Effective treatment of infectious diseases requires prompt and accurate bacterial identification and tailored antimicrobial treatments. Traditional culture methods are considered the gold standard, but their effectiveness diminishes for fastidious and hard-to-grow microorganisms. In recent years, molecular diagnostic tools such as 16S rRNA gene next-generation sequencing (16S NGS) have gained popularity in the field. We analysed data from samples submitted for 16S NGS between July 2022 and July 2023 at the Department of Advanced Translational Microbiology in Trieste, Italy. The study included samples submitted for both culture-based identification and 16S NGS. Conventional media were used for culture, and bacterial identification was performed using MALDI-TOF mass spectrometry. The V3 region of the 16S rRNA gene was sequenced using the Ion PGM platform. Among the 123 samples submitted, drainage fluids (38%) and blood (23%) were the most common, with requests predominantly from the Infectious Diseases (31.7%) and Orthopedic (21.13%) Units. In samples collected from patients with confirmed infections, 16S NGS demonstrated diagnostic utility in over 60% of cases, either by confirming culture results in 21% or providing enhanced detection in 40% of instances. Among the 71 patients who had received antibiotic therapies before sampling (mean 2.3 prior antibiotic days), pre-sampling antibiotic consumption did not significantly affect the sensitivity of 16S NGS. In routine microbiology laboratories, combining 16S NGS with culture method enhances the sensitivity of microbiological diagnostics, even when sampling is conducted during antibiotic therapy.

## 1. Introduction

Bacterial identification is of utmost importance in the treatment of infectious diseases. Early identification of the etiological agents is the key to personalised antimicrobial therapy as recently encouraged by antimicrobial stewardship programs. Although the traditional microbiological methods (microscopic examination and culturing) remain the gold standard for microbiological identification [1], their utility drastically decreases in cases of difficult-to-grow microorganisms or when patients are undergoing empirical antibiotic therapy [2]. Although matrix-assisted laser desorption/ionization–time of flight mass spectrometry (MALDI-TOF MS) has shortened the time required for bacterial identification, cultivated isolates are still essential for accurate identification [3]. Over the past few years, the popularity of next-generation sequencing (NGS) has increased among healthcare specialists. This diagnostic method has proved to be of ultimate importance in microbiota research, due to its capacity to identify bacterial diversity thanks to its ability to investigate any query in DNA clinical activity [4].

The accuracy of the 16S rRNA gene NGS (16S NGS) in analysing complex bacterial populations has made it a key instrument in diagnostic microbiology. This technique outperformed the traditional methods in cases of polymicrobial infections. Moreover, because of its capacity to detect low levels of bacterial DNA in clinical samples, 16S NGS has proved a powerful tool in the identification of difficult-to-grow bacteria. These advantages offer more accurate and comprehensive information to healthcare professionals, leading to more informed decisions and better management of patients in several settings [5,6,7], such as periprosthetic infections [8], osteomyelitis [9], and endocarditis [10]. In this regard, the latest version of the guidelines from the European Society of Cardiology recommends the use of molecular biology techniques to improve the diagnosis of culture-negative endocarditis [11].

Recent technical advancements in sequencing methods including short durations and low costs have increased their application in clinical microbiological settings, although in small-scale analysis, the cost per sample appears to still be high. Moreover, because of its high risk of contamination, 16S NGS results can be hard to analyse and require specialised expertise and laboratory equipment to provide the best results [12,13,14].

The aim of our study was to analyse the diagnostic power of 16S NGS in routine clinical samples, comparing it with traditional microbiological identification.

## 2. Materials and Methods

This study was designed as a retrospective observational study. We analysed the reports of all samples submitted for 16S NGS between July 2022 and July 2023 to the Department of Advanced Translational Microbiology, part of the Institute for Maternal and Child Health, IRCCS ”Burlo Garofolo”, Trieste, Italy. We recorded the results of the bacterial identification, the type of sample that was analysed and the suspected diagnosis.

We included in this study only the samples that were submitted for both culture method identification and 16S NGS. Data regarding the age and gender of the patients were anonymously recorded for each sample. Moreover, we collected information about the requesting department and the antibiotic therapies administered to these patients. The infection status was recorded from the patients’ files, based on all the clinical and laboratory findings for these patients during the current admission.

Conventional media were used for culture, and bacterial identification was performed using MALDI-TOF mass spectrometry (bioMérieux, Marcy-l’Etoile, France) The V3 region of 16S rRNA gene was sequenced using the Ion PGM platform (Thermo Fisher Scientific, Waltham, MA, USA).

All data were stored in a Microsoft Excel (Microsoft 365,Microsoft, Redmond, WA, USA) spreadsheet. For the statistical analysis, we used the IBM SPSS Statistics Version 22.0. (IBM, Armonk, NY, USA). Sensitivity (Se) and specificity (Sp) of the tests were calculated using the final diagnosis of the patients as the gold standard. The comparison of the groups was performed using Fischer’s exact test or chi-square test based on the sample size. A threshold of 0.05 was set for the *p*-values and any result < 0.05 was considered statistically significant.

The study was approved by the Ethics Committee of Trieste University n°V137_2201_24, in agreement with the Declaration of Helsinki (1964) and its later amendments.

## 3. Results

### 3.1. Descriptive Analysis

During the one-year period of our study, 154 requests were submitted for the 16S NGS examination, covering 135 patients. Samples for which the culture method was not performed were excluded from the database. Ultimately, the analysis was conducted on 123 independent samples.

The study group consisted of 123 patients, 75 of whom were male (60.97%); their ages ranged from 21 to 89 years (median = 67, Q1 = 53.5, Q3 = 75.5). The ages of the female patients (n = 48, 39.03%) ranged from 19 to 88 years (median = 69.5, Q1 = 52.75, Q3 = 77.5). The highest number of requests for 16S NGS examination came from the Infectious Diseases department (n = 39; 31.7%), with 15 of these samples being blood samples. Detailed representations of sample distributions are presented in Table 1 (infection sites) and Figure 1 (departments).

Out of the 123 samples collected, 36 (29.26%) were confirmed as culture positive, while in the NGS group, 71 (57.72%) samples were confirmed as harbouring at least one pathogen. Twenty-nine (23.57%) samples were confirmed as positive by both the culture method and NGS examination. Table 2 presents the comparison between the results of 16S NGS and the culture method. The culture method identified 32 out of 36 samples as monomicrobial, while NGS confirmed 38 out of the 71 positive results as monomicrobial. Only 11.11% (n = 4) of the culture results were confirmed as polymicrobial, while 46.47% (n = 33) of the NGS results were polymicrobial.

### 3.2. Diagnostic Tool Assessment

Out of the 123 samples collected, 99 (80.48%) samples were collected from patients with a confirmed infection. The sensitivity and specificity of the two diagnostic tools can be found in Table 3.

The results of the culture method and 16S NGS were concordant in 67 cases (54.47%). The concordance rate for culture-negative samples was 62.50%, while for culture-positive samples, it was 61.11%. A detailed presentation of the cases with discordant results is shown in Table 4. In 7 out of the 42 cases of culture-negative but 16S-positive results (3 cases of infective endocarditis and 4 bone infections), the blood cultures were positive and decisive for the diagnosis.

In samples collected from patients with a confirmed infection, 16S NGS was useful for diagnosis by either confirming culture results in 21.21% of the cases or providing enhanced detection in 40.40% of the cases. A more detailed view of the utility of 16S NGS is presented in Figure 2.

### 3.3. Diagnostic Tools Assessment

Prior to sampling, 71 patients received antibiotic therapies. The average number of days antibiotic therapy before sampling was 2.38 days, ranging from 1 to 14 days. The impact of antibiotic consumption on microbiology assessment results is presented in Figure 3. True positives were defined as cases for which the final diagnosis was an infectious pathology and the microbiological assessment could identify at least one pathogen. True negatives followed the same rule but applied to cases in which the final diagnosis was not an infectious pathology and microbiological assessment did not identify any bacteria. Antibiotic use prior to sampling had no statistically significant impact on the accuracy of 16S NGS (*p* > 0.05). However, antibiotic use reduced diagnostic sensitivity in samples subjected to the culture method.

In cases where both the culture and 16S NGS results were positive, but the pathogens identified were different, five out of seven samples were collected from patients who had previously received antibiotics. A detailed presentation of these cases is shown in Table 5.

## 4. Discussion

Ever since it was introduced in clinical practice, 16S NGS has proven to be an extraordinary diagnostic tool. However, like many other molecular techniques, it has some limitations, such as not being able to discriminate between viable and non-viable microorganisms or evaluate clinical phenotypic antimicrobial susceptibility. This necessitates close collaboration between the clinical practitioner and the microbiologist to analyse the results on a case-by-case basis.

In this study, we planned to evaluate the clinical relevance of 16S NGS and compare it with the culture method. Out of the 123 samples included in this study, the vast majority were drainage fluids and blood samples. Drainage fluids appear to be a common sample used for 16S NGS analysis, as previously described by Flurin et al., highlighting the NGS clinical value in terms of technical performance and timely diagnostic response [15]. In their paper, 40% of the samples were fluid samples, with the highest positivity rates of 16S NGS in abscess fluids.

The best accuracy of 16S NGS in our study was found in heart valve biopsies. Although the number of heart valve samples was small, 16S NGS correctly diagnosed 100% of these patients. In the patients for which both blood and valve cultures were performed, blood cultures returned positive results in only 50% of cases, while valve cultures had a detection rate of 0%. As previously described by Hong et al., in patients with suspected infective endocarditis, 16S NGS provided superior identification rates compared to the conventional blood cultures [16]. These results confirm the findings in our study. In addition, 16S NGS accurately diagnosed 64.70% of suspected infective endocarditis cases, whereas blood cultures yielded positive results in only 52.94%. It is important to consider the results of blood cultures and the antibiotic therapies prior to sampling to choose the proper assessment. Indeed, 16S NGS proved efficient even in cases of antibiotic administration and negative blood cultures. These results highlight the importance of 16S NGS in diagnosing bloodstream infections, especially when heart valve biopsies were collected.

Since prior antibiotic use can generally affect the microbiology results, we further analysed prior antibiotic use of the patients included in the study. We found that 71 patients received prior antibiotic treatment, and this had an impact on the results. Even 1–2 days of antibiotic treatment increased the false results of the culture method, while the 16S NGS results were only slightly affected. This trend remains similar in patients with more than 5 days of antibiotic prior to sampling. Our results clearly suggest that in patients with prior antibiotic treatment, 16S NGS might provide a better diagnosis and thus in this category of patients, this diagnostic tool should have priority [17,18].

Further analysing the cases with positive but discordant results, we noticed that out of the seven cases, five had previously received antibiotic treatment. This shows that in some cases, the results of 16S NGS might be difficult to interpret in a clinical scenario. Thus, we analyse the results in a case-by-case approach. In the first four cases of positive, but discordant results, the sample is represented by the drainage fluid. Prior antibiotic treatment might be the answer for this discordance for the first three cases. However, in case number four, the culture results showed evidence of *Staphylococcus aureus*, while 16S NGS showed *Massilia timonae*. *M. timonae* is a non-fermentative aerobic Gram-negative rod that was described previously in the literature as a human pathogen detected on very rare occasions. Thus, in that situation, probable contamination may pose a confusion factor since the bacterium can be considered an environmental organism [19]. A similar situation was reported in the following two cases where the samples were blood samples. In the only case where discordance was present in a bone biopsy sample, the final diagnosis was bone tuberculosis and the result provided by the 16S NGS showed *Stenotrophomonas maltophilia*. This bacterium is resistant to a wide range of antibiotics, and this might explain the selection of this bacterium and the result.

Regarding the sensitivity and specificity of the techniques, 16S NGS offers higher sensitivity than cultures with 68.69% compared to 36.36%. Similar data were provided by Abayasekara et al. who reported a sensitivity of 52.7%. However, our concordance was lower with only 61.11% compared with 91.8% for positive cultures [20]. In general, NGS has the advantage of high sensitivity and a wide detection range, but its specificity seems related to both the type of biological samples and the correct modality of sample collection to avoid contamination [21,22,23]. Moreover, additional comparative studies on large numbers of samples that consider patient symptoms, cure rates, and antibiotic selection are needed for better use in clinical routine. Using the culture method and 16S NGS has proven to be an effective method of increasing the diagnostic value of almost all the samples. As previously mentioned, in more than a fifth of the cases, 16S NGS confirmed the culture results, and in almost half of the situations, it provided better detection. Confirming the recently published data, in our series, the NGS results proved to be inconclusive from the clinical point of view for some samples, including bone, blood, and drainage fluids [23].

In addition, our data evidenced that for patients who underwent antibiotic therapies before sampling, it is important to consider the period of antibiotic administration. As previously stated by Lamoureux et al., antibiotic administration significantly diminishes the diagnostic power of culture methods, so molecular diagnostic tools are advised [24]. Although not statistically significant, the results of our study showed a trend for 16S NGS to provide better results if the sample is collected within 1–2 days of antibiotic therapy. The diagnostic power of 16S NGS after 2 days of antibiotic treatment is still uncertain and further studies should be considered.

Nonetheless, although 16S NGS is highly capable of identifying fastidious and slow-growing organisms, it is important to recall that all the results are connected to bioinformatics pipelines that are still under development [25]. As previously stated by Bores et al., these bioinformatics algorithms can sometimes influence the quality filtering and sequence classification [26]. Careful analysis and interpretation of the results are recommended for correlating clinical and epidemiological data in order to enhance the process of clinical decision-making.

Despite our data being monocentric and originating only from Italy, the relatively novel nature of the 16S NGS methodology we employed suggests that our paper may still hold global relevance. Although our results provide valuable insights in line with other recent studies, it is important to consider the limitations associated with this study. The relatively small sample size (n = 123) limits the generalisation of our findings, considering that this study could not capture the full spectrum of bacterial diversity and types of biological samples encountered in routine clinical practice.

## 5. Conclusions

The 16S NGS molecular technique is a new and effective alternative for bacterial identification. Based on our results, it may provide better results than the culture method in patients with prior antibiotic exposure. In over 60% of the cases, 16S NGS proved beneficial to clinical practice by confirming the culture results in 21.21% of cases or by offering enhanced detection in 40.40% of cases. Although 16S NGS shows great promise in diagnosing bacterial infections, the culture method remains vital for conducting antimicrobial susceptibility tests and is indispensable in cases where 16S NGS yields negative or polymicrobial results. Coupling 16S NGS with the culture method might be a promising alternative for providing rapid and accurate diagnoses in selected infectious diseases.

## Figures and Tables

**Figure 1 diagnostics-14-01318-f001:**
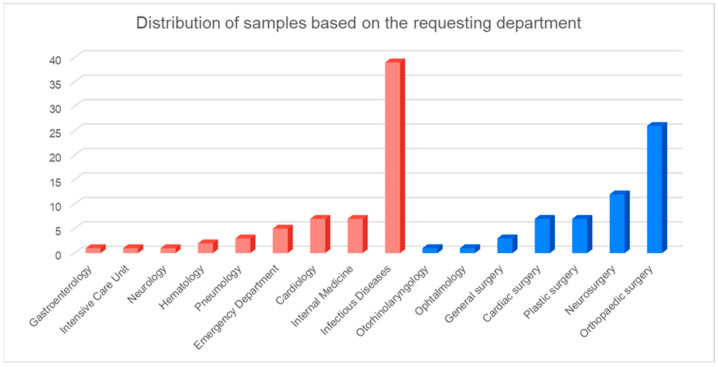
Surgical (blue) and medical departments (red) requesting 16S NGS examination.

**Figure 2 diagnostics-14-01318-f002:**
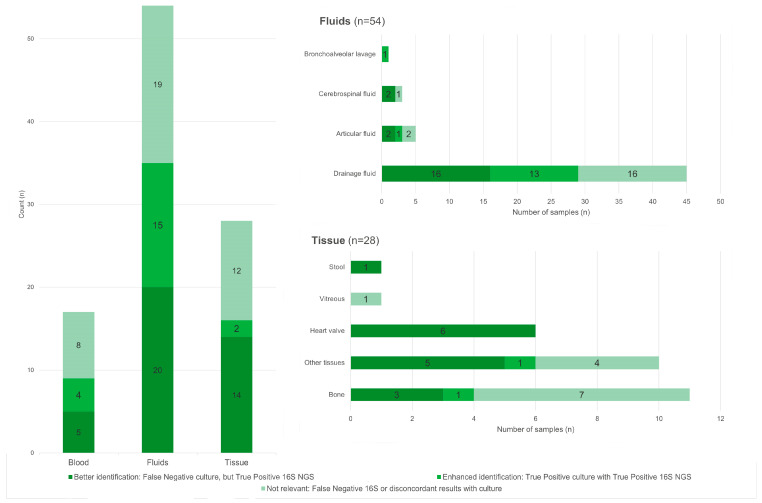
Impact of 16S NGS results in samples from patients with a confirmed infection.

**Figure 3 diagnostics-14-01318-f003:**
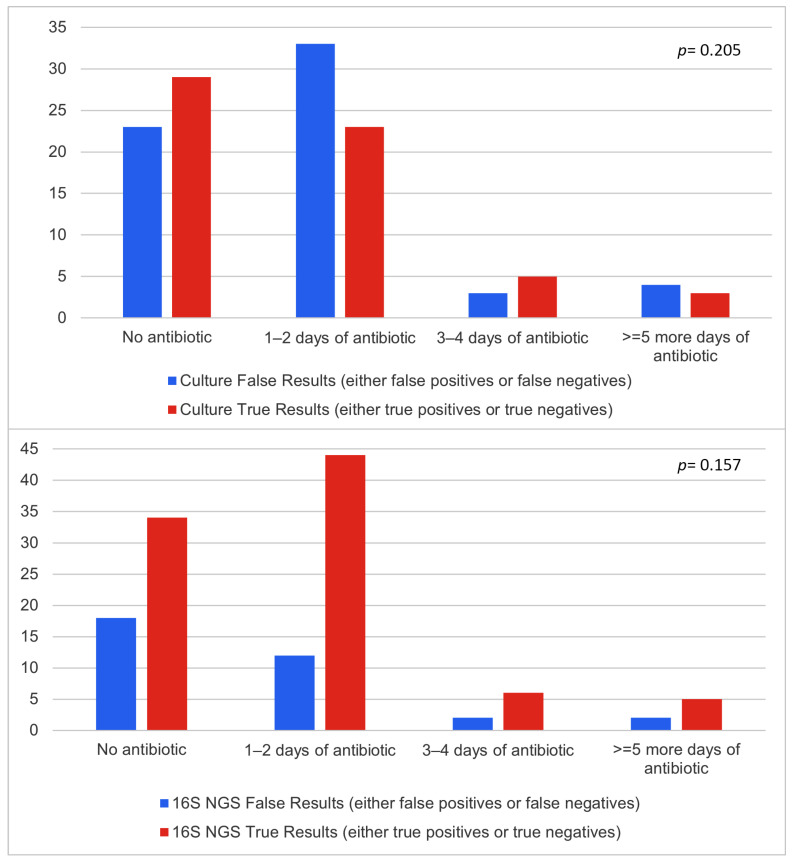
Effects of antibiotic administration on culture and 16S NGS results.

**Table 1 diagnostics-14-01318-t001:** Distribution of samples included in this study.

Samples	Number of Samples (%)
Solid samples	
Bone	11 (9%)
Heart valve	6 (5%)
Vitreous	1 (1%)
Stool	1 (1%)
Other tissues	11 (9%)
Fluid samples	
Drainage fluids	47 (38%)
Periprosthetic infection	12 (10%)
Osteomyelitis	10 (8%)
Intraabdominal infection	10 (8%)
Breast implant infection	7 (6%)
Pleural fluid	3 (2%)
Pericardic fluid	2 (2%)
Blood	29 (24%)
Articular fluid	9 (7%)
Cerebrospinal fluid	7 (6%)
Bronchoalveolar lavage	1 (1%)
Other fluids	3 (2%)

**Table 2 diagnostics-14-01318-t002:** Comparison of NGS results and culture method results for the samples analysed.

	Culture Negative	Culture Positive	Total
16S NGS negative	45	7	52
16S NGS positive	42	29	71
Total	87	36	123

**Table 3 diagnostics-14-01318-t003:** Diagnostic accuracy analysis for culture method and 16S NGS for the samples in this study.

	Infection Present	No Infection	*p*	Se	Sp
Culture positive	36	0	<0.001	36.36% 95% CI: [26.93; 46.64]	100% 95% CI: [85.75; 100]
Culture negative	63	24			
16S NGS positive	68	3	<0.001	68.69% 95% CI: [58.59; 77.64]	87.50% 95% CI: [67.64; 97.34]
16S NGS negative	31	21			

Abbreviations: CI, confidence interval, Se, sensitivity; Sp, specificity.

**Table 4 diagnostics-14-01318-t004:** Overview on discordant results between culture method and 16S NGS.

	Number of Cases	Samples
Culture-positive 16S NGS-negative results	7	Blood sample (n = 3, 6.38%) Drainage fluids (n = 2, 4.25%) Articular fluid (n = 1, 11.11%) Other tissue (n = 1, 9.09%)
Culture-negative 16S NGS-positive results	42 *	Drainage liquid (n = 15, 31.19%) Heart valve (n = 6, 100%) Blood sample (n = 6, 20.68%) Other tissue (n = 6, 54.54%) Bone (n = 3, 27.27%) Cerebrospinal fluid (n = 3, 42.85%) Articular fluid (n = 2, 22.22%) Stool sample (n = 1, 100%)
Culture-positive 16S NGS-positive results, but different pathogens identified	7	Drainage fluid (n = 4, 8.51%) Blood sample (n = 2, 6.89%) Bone (n = 1, 9.09%)

* Blood cultures confirmed the 16S NGS results for 7 cases (3 heart valves, 2 other tissues, 2 bone) in this group.

**Table 5 diagnostics-14-01318-t005:** Overview of cases in which the culture method and 16S NGS identified different pathogens.

	Sample	Antibiotic Days	Antibiotic Used	Culture Result	16S NGS Result
1	Drainage fluid	2 days	LVX/RIF	*Corynebacterium striatum*	*Enterococcus* spp./ *Cutibacterium acnes*/ *Acinetobacter* spp./ *Bosea vestrisii*
2	Drainage fluid	2 days	VAN/PIP/TAZ	*Enterococcus* *faecium*	*Corynebacterium tuberculostearicum*
3	Drainage fluid	2 days	AMC	*Klebsiella oxytoca* *Citrobacter braakii*	*Citrobacter freundii*/ *Proteus vulgaris*/ *Veillonella parvula*/ *Morganella morganii*/ *Enterobacter* spp./ *Enterococcus faecalis*/ *Streptococcus pneumoniae*/ *Aeromonas veronii*/ *Escherichia coli*/ *Clostridium perfringens*
4	Drainage fluid	No antibiotic	-	*Staphylococcus* *aureus*	*Massilia timonae*
5	Blood	3 days	LVX	*Staphylococcus* *aureus*	*Cutibacterium acnes* *Serratia marcescens*
6	Blood	No antibiotic	-	*Escherichia coli*	*Acinetobacter junii*/ *Lactobacillus crispatus*
7	Bone	14 days	ETA/Rifater	*M. tuberculosis complex*	*Stenotrophomonas maltophilia*

Abbreviations: AMC, amoxicillin/clavulanic acid; ETA, ethambutol; LVX, levofloxacin; PIP, piperacillin; RIF, rifampicin; TAZ, tazobactam; VAN, vancomycin.

## Data Availability

The datasets analysed during the current study are not publicly available but are available from the corresponding authors upon reasonable request.

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
