# Peer review of "Performance of 16S rRNA Gene Next-Generation Sequencing and the Culture Method in the Detection of Bacteria in Clinical Specimens"

_diagnostics, 2024, doi:10.3390/diagnostics14131318_

Round 1

Reviewer 1 Report

Comments and Suggestions for Authors

The article "Performance of the 16S rRNA Gene Next-Generation Sequencing and the Culture Method in the Detection of Bacteria in Clinical Specimens" addresses a current topic regarding etiological diagnostic methods in the case of infections. It is a well-executed article.

Recommendations:

    1. The introduction is quite short and could be improved. The purpose of the article is not presented.
    2. The results presented in Figure 1 are difficult to follow; I suggest changing the type of figure or replacing it with a table. In the results section, it would be useful to present the species of bacteria and their prevalence in the samples studied.
    3. The bibliography should be improved.

Author Response

“1. The introduction is quite short and could be improved. The purpose of the article is not presented.”

We have extended the introduction and included a sentence with the purpose of the study.

“2. The results presented in Figure 1 are difficult to follow; I suggest changing the type of figure or replacing it with a table. In the results section, it would be useful to present the species of bacteria and their prevalence in the samples studied.”

Thank you for your suggestion. Figure 1 was replaced with Table 1. Unfortunately, because of the high variety of bacteria in our samples, we are not able to contain the prevalence in a single table/figure. Moreover, no correlation between samples and bacteria was significant.

“3. The bibliography should be improved.”

We added more references to the bibliography, as suggested.

Reviewer 2 Report

Comments and Suggestions for Authors

This paper reviews Next-Generation Sequencing (NGS) for its application as a Clinical Specimen. The manuscript is meticulously and rationally categorized, and since the research period is recent, it can be considered state-of-the-art. The overall structure is sound, and the conclusions drawn from the content are comprehensible. However, this reviewer raises several essential questions that need to be addressed:

1. Explain whether the research content limited to Italy holds global relevance.

2. Is the sample size of 123 adequate? If so, provide justification for its sufficiency. Additionally, explain the reliability of results if the sample size were increased.

3. Were the samples without antibiotic history separated from those with prior antibiotic administration? Does the individual treatment history affect the results?

4. Is a detection rate of 40% sufficiently high? Is the remaining 60% unpredictable, and if so, is it still useful?

minor. The colors in the figures are faint. They should be made more vivid.

Author Response

“1. Explain whether the research content limited to Italy holds global relevance.”

We added a sentence in the final part of Discussion. Thank you for your suggestion. 

“2. Is the sample size of 123 adequate? If so, provide justification for its sufficiency. Additionally, explain the reliability of results if the sample size were increased.”

Thank you for your comment. Considering the retrospective and single-center nature of the study, we believe that our sample size is adequate. 16S NGS is not widely available, and therefore, there are considerable limitations regarding the potential increase in sample size. We must mention the increased cost of this methodology and the difficulties in data analysis. All of these points have already been discussed in the paper. 

“3. Were the samples without antibiotic history separated from those with prior antibiotic administration? Does the individual treatment history affect the results?”

Yes, in paragraph 3.2 of the “Results” section we analysed patients according to prior antibiotic treatment and we presented the impact of antibiotic consumption on microbiology assessment results in Figure 3. 

“4. Is a detection rate of 40% sufficiently high? Is the remaining 60% unpredictable, and if so, is it still useful?”

We believe that 40% detection rate actually provides answers for specific cases in which conventional methods failed to provide diagnosis. Therefore, in a clinical setting it may prove sufficiently high. We can translate these results in 40% more cases in which patient management was positively influenced. In addition to the 21.21% in which 16S NGS confirmed the culture method, we consider that further studies should be conducted in order to provide specific guidelines for each sample type for the best accuracy and cost efficiency.

“5. minor. The colors in the figures are faint. They should be made more vivid.”

Thank you for your suggestion. We increased the saturation of the figures to 200% in order to be more vibrant.
